# Direct Multi-view Multi-person 3D Pose Estimation

**Tao Wang**[1,2]*, **Jianfeng Zhang**[2]*, **Yujun Cai**[1], **Shuicheng Yan**[1], **Jiashi Feng**[1],
[1]Sea AI Lab [2]National University of Singapore,
twangnh@gmail.com,
zhangjianfeng@u.nus.edu,
{caiyj,yansc,fengjs}@sea.com

## Abstract

We present **M**ulti-**v**iew **P**ose transformer (MvP) for estimating multi-person 3D poses from multi-view images. Instead of estimating 3D joint locations from costly volumetric representation or reconstructing the per-person 3D pose from multiple detected 2D poses as in previous methods, MvP directly regresses the multi-person 3D poses in a clean and efficient way, without relying on intermediate tasks. Specifically, MvP represents skeleton joints as learnable query embeddings and let them progressively attend to and reason over the multi-view information from the input images to directly regress the actual 3D joint locations. To improve the accuracy of such a simple pipeline, MvP presents a hierarchical scheme to concisely represent query embeddings of multi-person skeleton joints and introduces an input-dependent query adaptation approach. Further, MvP designs a novel geometrically guided attention mechanism, called *projective attention*, to more precisely fuse the cross-view information for each joint. MvP also introduces a RayConv operation to integrate the view-dependent camera geometry into the feature representations for augmenting the projective attention. We show experimentally that our MvP model outperforms the state-of-the-art methods on several benchmarks while being much more efficient. Notably, it achieves 92.3% $AP_{25}$ on the challenging Panoptic dataset, improving upon the previous best approach [40] by 9.8%. MvP is general and also extendable to recovering human mesh represented by the SMPL model, thus useful for modeling multi-person body shapes. Code and models are available at https://github.com/sail-sg/mvp.

## 1 Introduction

Multi-view multi-person 3D pose estimation aims to localize 3D skeleton joints for each person instance in a scene from multi-view camera inputs. It is a fundamental task that benefits many real-world applications (such as surveillance, sportscast, gaming and mixed reality) and is mainly tackled by reconstruction-based [6, 14, 4] and volumetric [40] approaches in previous literature, as shown in Fig. 1 (a) and (b). The former first estimates 2D poses in each view independently and then aggregates them and reconstructs their 3D counterparts via triangulation or a 3D pictorial structure model. The volumetric approach [40] builds a 3D feature volume through heatmap estimation and 2D-to-3D un-projection at first, based on which instance localization and 3D pose estimation are performed for each person instance individually. Though with notable accuracy, the above paradigms are inefficient due to highly relying on those intermediate tasks. Moreover, they estimate 3D pose for each person separately, making the computation cost grow linearly with the number of persons.

Targeted at a more simplified and efficient pipeline, we were wondering if it is possible to *directly* regress 3D poses from multi-view images without relying on any intermediate task? Though conceptually attractive, adopting such a direct mapping paradigm is highly non-trivial as it remains

---
*Equal Contribution.

35th Conference on Neural Information Processing Systems (NeurIPS 2021).

unclear how to perform skeleton joints detection and association for multiple persons within a single stage. In this work, we address these challenges by developing a novel **M**ulti-**v**iew **P**ose transformer (MvP) model which significantly simplifies the multi-person 3D pose estimation. Specifically, MvP represents each skeleton joint as a learnable positional embedding, named *joint query*, which is fed into the model and mapped into final 3D pose estimation directly (Fig. 1 (c)), via a specifically designed attention mechanism to fuse multi-view information and globally reason over the joint predictions to assign them to the corresponding person instances. We develop a novel hierarchical query embedding scheme to represent the multi-person joint queries. It shares joint embedding across different persons and introduces person-level query embedding to help the model in learning both person-level and joint-level priors. Benefiting from exploiting the person-joint relation, the model can more accurately localize the 3D joints. Further, we propose to update the joint queries with input-dependent scene-level information (*i.e.*, globally pooled image features from multi-view inputs) such that the learnt joint queries can adapt to the target scene with better generalization performance.

To effectively fuse the multi-view information, we propose a geometrically-guided projective attention mechanism. Instead of applying full attention to densely aggregate features across spaces and views, it projects the estimated 3D joint into 2D anchor points for different views, and then selectively fuses the multi-view local features near to these anchors to precisely refine the 3D joint location. we propose to encode the camera rays into the multi-view feature representations via a novel RayConv operation to integrate multi-view positional information into the projective attention. In this way, the strong multi-view geometrical priors can be exploited by projective attention to obtain more accurate 3D pose estimation.

Comprehensive experiments on 3D pose benchmarks Panoptic [19], as well as Shelf and Campus [1] demonstrate our MvP works very well. Notably, it obtains 92.3% $AP_{25}$ on the challenging Panoptic dataset, improving upon the previous best approach VoxelPose [40] by 9.8%, while achieving nearly $2\times$ speed up. Moreover, the design ethos of our MvP can be easily extended to more complex tasks—we show that a simple body mesh branch with SMPL representation [28] trained on top of a pre-trained MvP can achieve competitively qualitative results.

Our contributions are summarized as follows: 1) We strive for simplicity in addressing the challenging multi-view multi-person 3D pose estimation problem by casting it as a **direct regression problem** and accordingly develop a novel Multi-view Pose transformer (MvP) model, which achieves state-of-the-art results on the challenging Panoptic benchmark. 2) Different from query embedding designs in most transformer models, we propose a more tailored and concise hierarchical joint query embedding scheme to enable the model to effectively encode person-joint relation. Additionally, we mitigate the commonly faced generalization issue by a simple query adaptation strategy. 3) We propose a novel projective attention module along with a RayConv operation for fusing multi-view information effectively, which we believe are also inspiring for model designs in other multi-view 3D tasks.

## 2    Related Works

**3D Human Pose Estimation**    3D pose estimation from monocular inputs [29, 30, 49, 35, 38, 31, 46, 10, 47] is an ill-posed problem as multiple 3D predictions may result in the same 2D projection. To alleviate such projective ambiguities, multi-view methods have been explored. Research works on single-person scenes use either multi-view geometry [11] for feature fusion [36, 13] and triangulation [16, 37], or pictorial structure models for fast and robust 3D pose reconstruction [34, 36], achieving promising results. However, it is more challenging as we progress towards multi-person scenes. Current approaches mainly exploit a multi-stage pipeline for multi-person tasks, including reconstruction-based [6, 4, 14, 21, 26] and volumetric [40] paradigms. Despite their notable accuracy, these methods suffer expensive computation cost from the intermediate tasks, such as cross-view matching and heatmap back-projection. Moreover, the total computation cost grows linearly with the number of persons in the scene, making them hardly scalable for larger scenes. Different from all previous approaches that rely on a multi-stage pipeline with computation redundancy, our method views multi-person 3D pose estimation as a **direct regression problem** based on a novel Multi-view Pose transformer model, enables an intermediate task-free single stage solution.

**Attention and Transformers**    Driven by the recent success in natural language fields, there have been growing interests in exploring the Transformers for computer vision tasks, such as image recognition [8] and generation [18], as well as more complicated object detection [3, 51] and video

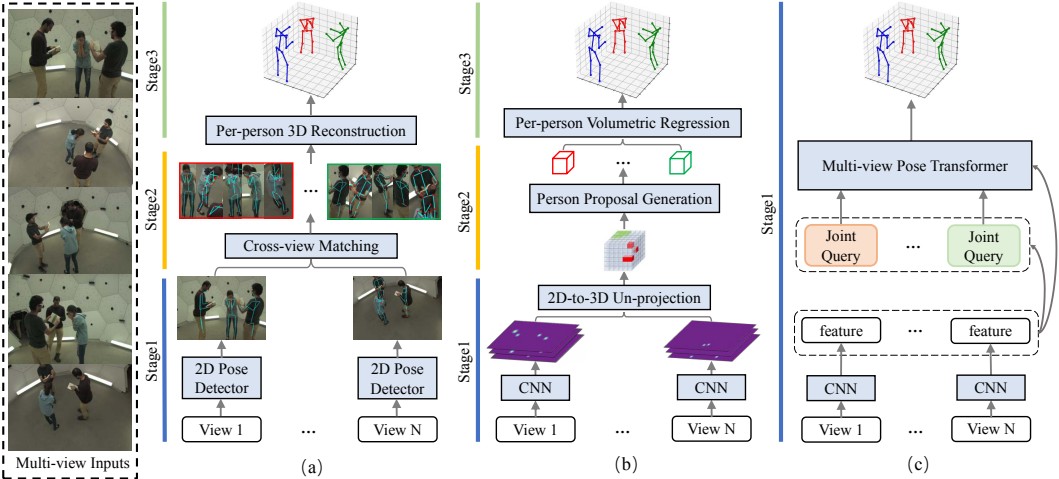

Figure 1: Difference between our method and others for multi-view multi-person 3D pose estimation. Existing methods adopt complex multi-stage pipelines that are either (a) reconstruction-based or (b) volumetric representation based, which incur heavy computation burden. (c) Our method solves this task as a **direct regression problem** without relying on any intermediate task by a novel Multi-view Pose Transformer, and largely simplifies the pipeline and boosts the efficiency.

instance segmentation [42]. However, multi-person 3D pose estimation has not been explored along this direction. In this study, we propose a novel Multi-view Pose Transformer architecture with a joint query embedding scheme and a projective attention module to regress 3D skeleton joints from multi-view images directly, delivering a simplified and effective pipeline.

## 3   Multi-view Pose Transformer (MvP)

To build a direct multi-person 3D pose estimation framework from multi-view images, we introduce a novel **M**ulti-**v**iew **P**ose transformer (MvP). MvP takes in the multi-view feature representations, and transforms them into groups of 3D joint locations directly (Fig. 2 (a)), delivering multi-person 3D pose results, with the following carefully designed query embedding and attention schemes for detecting and grouping the skeleton joints.

### 3.1   Joint Query Embedding Scheme

Inspired by transformers [41], MvP represents each skeleton joint as a learnable positional embedding, which is fed into the transformer decoder and mapped into final 3D joint location by jointly attending to other joints and the multi-view information (Fig. 2 (a)). The learnt embeddings encode *a prior* knowledge about the skeleton joints and we name them as *joint queries*. MvP develops the following concise query embedding scheme.

**Hierarchical Query Embeddings**   The most straightforward way for designing joint query embeddings is to maintain a learnable query vector for each joint per person. However, we empirically find this scheme does not work well, likely because such a naive strategy cannot share the joint-level knowledge between different persons.

To tackle this problem, we develop a hierarchical query embedding scheme to explicitly encode the person-joint relation for better generalization to different scenes. The hierarchical embedding offers joint-level information sharing across different persons and reduces the learnable parameters, helping the model to learn useful knowledge from the training data, and thus generalize better. Concretely, instead of using the set of independent joint queries $\{\mathbf{q}_m\}_{m=1}^{M} \subset \mathbb{R}^C$, we employ a set of person level queries $\{\mathbf{h}_n\}_{n=1}^{N} \subset \mathbb{R}^C$, and a set of joint level queries $\{\mathbf{l}_j\}_{j=1}^{J} \subset \mathbb{R}^C$ to represent different persons and different skeleton joints, where $C$ denotes the feature dimension, $N$ is the number of persons, $J$ is the number of joints per person, and $M = NJ$. Then the query of joint $j$ of person $n$

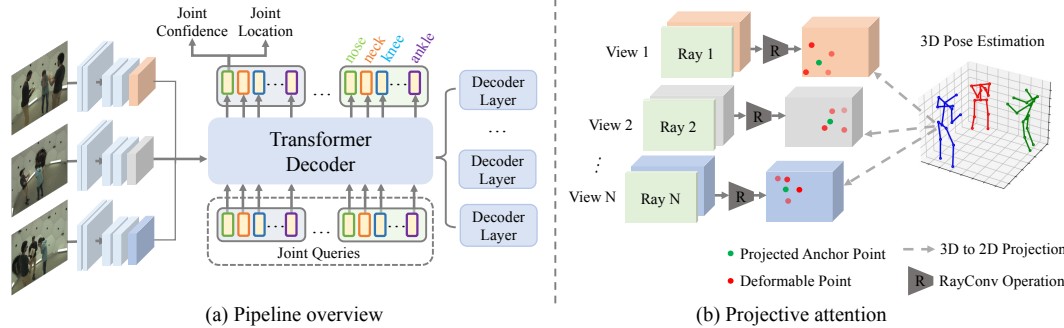

(a) Pipeline overview      (b) Projective attention

Figure 2: (a) Overview of the proposed MvP model. Upon the multi-view image features from several convolution layers, it deploys a transformer decoder with a stack of decoder layers to map the input joint queries and the multi-view features to 3D poses directly. (b) The projective attention of MvP projects 3D skeleton joints to anchor points (the green dots) on different views and samples deformable points (the red dots) surrounding these anchors to aggregate local contextual features via learned weights (the brighter color density means larger weights).

can be hierarchically formulated as

$$\mathbf{q}_n^j = \mathbf{h}_n + \mathbf{l}_j. \tag{1}$$

With such a hierarchical embedding scheme, the number of learnable query embedding parameters is reduced from $NJC$ to $(N + J)C$.

**Input-dependent Query Adaptation**    In the above, the learned joint query embeddings are shared for all the input images, independent of their contents, and thus may not generalize well on the novel target data. To address this limitation, we propose to augment the joint queries with input-dependent scene-level information in both model training and deployment, such that the learnt joint queries can be adaptive to the target data and generalize better. Concretely, we augment the above joint queries with a globally pooled feature vector $\mathbf{g} \in \mathbb{R}^C$ from the multi-view image feature representations:

$$\mathbf{q}_n^j = \mathbf{g} + \mathbf{h}_n + \mathbf{l}_j. \tag{2}$$

Here $\mathbf{g} = \mathrm{Concat}(\mathrm{Pool}(\mathbf{Z}_1), \ldots, \mathrm{Pool}(\mathbf{Z}_V))\mathbf{W}^g$, where $\mathbf{Z}_v$ denotes image feature from $v$-th view and $V$ is the total number of camera views; $\mathrm{Concat}$ and $\mathrm{Pool}$ denote concatenation and pooling operations, and $\mathbf{W}^g$ is a learnable linear weight.

### 3.2   Projective Attention for Multi-view Feature Fusion

It is crucial to aggregate complementary multi-view information to transform the joint embeddings into accurate 3D joint locations. We consider the dot product attention mechanism of transformers [41] to fuse the multi-view image features. However, naively applying such dot product attention densely over all spatial locations and camera views will incur enormous computation cost. Moreover, such dense attention is difficult to optimize and delivers poor performance empirically since it does not exploit any 3D geometric knowledge.

Therefore, we propose a geometrically-guided multi-view projective attention scheme, named projective attention. The core idea is to take the 2D projection of the estimated 3D joint location as the anchor point in each view, and only fuse the local features near those projected 2D locations from different views. Motivated by the deformable convolution [5, 50], we adopt an adaptive deformable sampling strategy to gather the localized context information in each camera view, as shown in Fig. 2 (b). Other local attention operations [48, 44, 43] can also be adopted as an alternative. Formally, given joint query feature $\mathbf{q}$ and 3D joint position $\mathbf{y}$, the projective attention is defined as

$$\mathrm{PAttention}(\mathbf{q}, \mathbf{y}, \{\mathbf{Z}_v\}_{v=1}^V) = \mathrm{Concat}(\mathbf{f}_1, \mathbf{f}_2, \ldots, \mathbf{f}_V)\mathbf{W}^P,$$
$$\text{where } \mathbf{f}_v = \sum_{k=1}^{K} \mathbf{a}(k) \cdot \mathbf{Z}_v\big(\Pi(\mathbf{y}, \mathbf{C}_v) + \Delta\mathbf{p}(k)\big)\mathbf{W}^f. \tag{3}$$

Here the view-specific feature $\mathbf{f}_v$ is obtained by aggregating features from $K$ discrete offsetted sampling points from an anchor point $\mathbf{p} = \Pi(\mathbf{y}, \mathbf{C}_v)$, located by projecting the current 3D joint

location $\mathbf{y}$ to 2D, where $\Pi : \mathbb{R}^3 \to \mathbb{R}^2$ denotes perspective projection [11] and $\mathbf{C}_v$ the corresponding camera parameters. $\mathbf{W}^P$ and $\mathbf{W}^f$ are learnable linear weights. The attention weight $\mathbf{a}$ and the offset to the projected anchor point $\Delta \mathbf{p}$ are estimated from the fusion of query feature $\mathbf{q}$ and the view-dependent feature at the projected anchor point $\mathbf{Z}_v(\mathbf{p})$, *i.e.*, $\mathbf{a} = \mathrm{Softmax}((\mathbf{q} + \mathbf{Z}_v(\mathbf{p}))\mathbf{W}^a)$ and $\Delta \mathbf{p} = (\mathbf{q} + \mathbf{Z}_v(\mathbf{p}))\mathbf{W}^p$, where $\mathbf{W}^a$ and $\mathbf{W}^p$ are learnable linear weights. If the projected location and the offset are fractional, we use bilinear interpolation to obtain the corresponding feature $\mathbf{Z}_v(\mathbf{p})$ or $\mathbf{Z}_v(\mathbf{p} + \Delta \mathbf{p}(t))$.

The projective attention incorporates two geometrical cues, *i.e.*, the corresponding 2D spatial locations across views from the 3D to 2D projection and the deformed neighborhood of the anchors from the learned offsets to gather view-adaptive contextual information. Unlike naive attention where the query feature densely interacts with the multi-view key features across all the spatial locations, the projective attention is more selective for the interaction between the query and each view—only the features from locations near to the projected anchors are aggregated, and thus is much more efficient.

**Encoding Multi-view Positional Information with RayConv** The positional encoding [41] is an important component of the transformer, which provides positional information of the input sequence. However, a simple per-view 2D positional encoding scheme cannot encode the multi-view geometrical information. To tackle this limitation, we propose to encode the camera ray directions that represent positional information in 3D space into the multi-view feature representations. Concretely, the camera ray direction $\mathbf{R}_v$, generated with the view-specific camera parameters, is concatenated channel-wisely to the corresponding image feature representation $\mathbf{Z}_v$. Then a standard convolution is applied to obtain the updated feature representation $\hat{\mathbf{Z}}_v$, with the view-dependent geometric information:

$$\hat{\mathbf{Z}}_v = \mathrm{Conv}(\mathrm{Concat}(\mathbf{Z}_v, \mathbf{R}_v)). \tag{4}$$

We name the operation as *RayConv*. With it, the obtained feature representation $\hat{\mathbf{Z}}_v$ is used for the projective attention by replacing $\mathbf{Z}_v$ in Eqn. (3).

Such drop-in replacement introduces negligible computation, while injecting strong multi-view geometrical prior to augment the projective attention scheme, thus helping more precisely predict the refined 3D joint position.

## 3.3 Architecture

Our overall architecture (Fig. 2 (a)) is pleasantly simple. It adopts a convolution neural network, designed for 2D pose estimation [45], to obtain high-resolution image features $\{\mathbf{Z}_v\}_{v=1}^V$ from multi-view inputs $\{\mathbf{I}_v\}_{v=1}^V$. The features are then fed into the transformer decoder consisting of multiple decoder layers to predict the 3D joint locations. Each layer conducts a self-attention to perform pair-wise interaction between all the joints from all the persons in the scene; a projective attention to selectively gather the complementary multi-view information; and a feed-forward regression to predict the 3D joint positions and their confidence scores. Specifically, the transformer decoder applies a *multi-layer progressive regression scheme*, *i.e.*, each decoder layer outputs 3D joint offsets to refine the input 3D joint positions from previous layer.

**Extending to Body Mesh Recovery** MvP learns skeleton joints feature representations and is extendable to recovering human mesh with a parametric body mesh model [28]. Specifically, after average pooling on the joint features into per-person feature, a feed-forward network is used to predict the corresponding body mesh represented by the parametric SMPL model [28]. Similar to the joint location prediction, the SMPL parameters follow multi-layer progressive regression scheme.

## 3.4 Training

MvP infers a fixed set of $M$ joint locations for $N$ different persons, where $M = NJ$. The main training challenge is how to associate the skeleton joints correctly for different person instances. Unlike the post-hoc grouping of detected skeleton joints as in bottom-up pose estimation methods [32, 24], MvP learns to directly predict the multi-joint 3D human pose in a group-wise fashion as shown in Fig. 2 (a). This is achieved by a grouped matching strategy during model training.

**Grouped Matching** Given the predicted joint positions $\{\mathbf{y}_m\}_{m=1}^M \subset \mathbb{R}^3$ and associated confidence scores $\{s_m\}_{m=1}^M$, we group every consecutive $J$-joint predictions into per-person pose estimation

$\{\mathbf{Y}_n\}_{n=1}^N \subset \mathbb{R}^{J \times 3}$, and average their corresponding confidence scores to obtain the per-person confidence scores $\{p_n\}_{n=1}^N$. The same grouping strategy is used during inference.

The ground truth set $\mathbf{Y}^*$ of 3D poses of different person instances is smaller than the prediction set of size $N$, which is padded to size $N$ with empty element $\varnothing$. Then we find a bipartite matching between the prediction set and the ground truth set by searching for a permutation of $\hat{\sigma} \in \aleph_N$ that achieves the lowest matching cost:

$$\hat{\sigma} = \arg\min_{\sigma \in \aleph_N} \sum_{n=1}^N \mathcal{L}_{\text{match}}(\mathbf{Y}_n^*, \mathbf{Y}_{\sigma(n)}). \tag{5}$$

We consider both the regressed 3D joint position and confidence score for the matching cost:

$$\mathcal{L}_{\text{match}}(\mathbf{Y}_n^*, \mathbf{Y}_{\sigma(n)}) = -p_i + \mathcal{L}_1(\mathbf{Y}_n^*, \mathbf{Y}_{\sigma(n)}) \tag{6}$$

where $\mathbf{Y}_n^* \neq \varnothing$, and $\mathcal{L}_1$ computes the $L_1$ loss error. Following [3, 39], we employ the Hungarian algorithm [25] to compute the optimal assignment $\hat{\sigma}$ with the above matching cost.

**Objective Function**   We compute the *Hungarian loss* with the obtained optimal assignment $\hat{\sigma}$:

$$\mathcal{L}_{\text{Hungarian}}(\mathbf{Y}^*, \mathbf{Y}) = \sum_{n=1}^N \left[ \mathcal{L}_{\text{conf}}(\mathbf{Y}_n^*, p_{\hat{\sigma}(n)}) + \mathbb{1}_{\{\mathbf{Y}_n^* \neq \varnothing\}} \lambda \mathcal{L}_{\text{pose}}(\mathbf{Y}_n^*, \mathbf{Y}_{\hat{\sigma}(n)}) \right]. \tag{7}$$

Here $\mathcal{L}_{\text{conf}}$ and $\mathcal{L}_{\text{pose}}$ are losses for confidence score and pose regression, respectively. $\lambda$ balances the two loss terms. We use focal loss [27] for confidence prediction which adaptively balances the positive and negative samples. For pose regression, we compute $L_1$ loss for 3D joints and their projected 2D joints in different views.

To learn multi-layer progressive regression, the above matching and loss are applied for each decoder layer. The total loss is thus $\mathcal{L}_{\text{total}} = \sum_{l=1}^L \mathcal{L}_{\text{Hungarian}}^l$, where $\mathcal{L}_{\text{Hungarian}}^l$ denotes loss of the $l$-th decoder layer and $L$ is the number of decoder layers. When extending MvP to body mesh recovery, we apply $L_1$ loss for 3D joints from the SMPL model and their 2D projections, as well as an adversarial loss following HMR [22, 17, 47] due to lack of GT SMPL parameters.

# 4   Experiments

In this section, we aim to answer following questions. 1) Can MvP provide both efficient and accurate multi-person 3D pose estimation? 2) How does the proposed attention mechanism help multi-view multi-person skeleton joints information fusing? 3) How does each individual design choice affect model performance? To this end, we conduct extensive experiments on several benchmark datasets.

**Datasets**   *Panoptic* [20] is a large-scale benchmark with 3D skeleton joint annotations. It captures daily social activities in an indoor environment. We conduct extensive experiments on Panoptic to evaluate and analyze our approach. Following VoxelPose [40], we use the same data sequences except '160906_band3' in the training set due to broken images. Unless otherwise stated, we use five HD cameras (3, 6, 12, 13, 23) in our experiments. All results reported in the experiments follow the same data setup. We use Average Precision (AP) and Recall [40], as well as Mean Per Joint Position Error (MPJPE) as evaluation metrics. *Shelf* and *Campus* [1] are two multi-person datasets capturing indoor and outdoor environments, respectively. We split them into training and testing sets following [1, 6, 40]. We report Percentage of Correct Parts (PCP) for these two datasets.

**Implementation Details**   Following VoxelPose [40], we adopt a pose estimation model [45] build upon ResNet-50 [12] for multi-view image features extraction. Unless otherwise stated, we use a

Table 1: Result on the Panoptic dataset. MvP is more accurate and faster than VoxelPose.

| Methods | $\text{AP}_{25}$ | $\text{AP}_{50}$ | $\text{AP}_{100}$ | $\text{AP}_{150}$ | Recall$_{@500}$ | MPJPE[mm] | Time[ms] |
|---|---|---|---|---|---|---|---|
| VoxelPose [40] | 84.0 | 96.4 | **97.5** | **97.8** | 98.1 | 17.8 | 320 |
| MvP (Ours) | **92.3** | **96.6** | **97.5** | 97.7 | **98.2** | **15.8** | **170** |

stack of six transformer decoder layers. The model is trained for 40 epochs, with the Adam optimizer of learning rate $10^{-4}$. During inference, a confidence threshold of 0.1 is used to filter out redundant predictions. Please refer to supplementary for more implementation details.

## 4.1 Main Results

**Panoptic**    We first evaluate our MvP model on the challenging Panoptic dataset and compare it with the state-of-the-art VoxelPose model [40]. As shown in Table 1, Our MvP achieves 92.3 $AP_{25}$, improving upon VoxelPose by 9.8%, and achieves much lower MPJPE (15.8 *v.s* 17.8). Moreover, MvP only requires 170ms to process a multi-view input, about $2\times$ faster than Voxel-Pose[2]. These results demonstrate both accuracy and efficiency advantages of MvP from estimating 3D poses of multiple persons in a direct regression paradigm.  To further demonstrate efficiency of MvP, we compare its inference time with VoxelPose's when processing different numbers of person instances. As shown in Fig. 3, the inference time of VoxelPose grows linearly with the number of persons in the scene due to the per-person regression paradigm. In

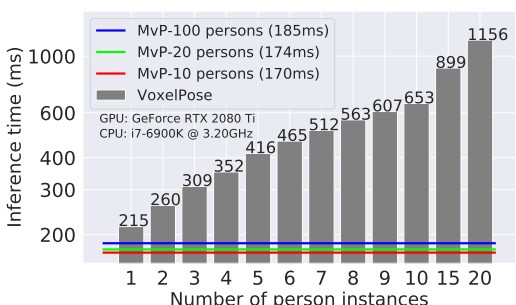

Figure 3: Inference time versus the number of person instances. Benefiting from its direct inference framework, MvP maintains almost constant inference time regardless of the number of persons.

contrast, MvP keeps constant inference time no matter how many instances in the scene. Notably, it takes only 185ms for MvP to process scenes even with 100 person instances (the blue line), demonstrating its great potential to handle crowded scenarios.

**Shelf and Campus**    We further compare our MvP with state-of-the-art approaches on the Shelf and Campus datasets. The reconstruction-based methods [2, 9, 6] use 3D pictorial model [2, 6] or conditional random field [9] within a multi-stage paradigm; and the volumetric approach VoxelPose [40] highly relies on computationally intensive intermediate tasks. As shown in Table 2, our MvP achieves the best performance in all the actors on the Shelf dataset. Moreover, it obtains a comparable result on the Campus dataset as VoxelPose [40] without relying on any intermediate task. These results further confirm the effectiveness of MvP for estimating 3D poses of multiple persons directly.

Table 2: Results (in PCP) on Shelf and Campus datasets.

| Methods | Shelf | | | | Campus | | | |
|---|---|---|---|---|---|---|---|---|
| | Actor 1 | Actor 2 | Actor 3 | Average | Actor 1 | Actor 2 | Actor 3 | Average |
| Belagiannis *et al.* [2] | 75.3 | 69.7 | 87.6 | 77.5 | 93.5 | 75.7 | 84.4 | 84.5 |
| Ershadi *et al.* [9] | 93.3 | 75.9 | 94.8 | 88.0 | 94.2 | 92.9 | 84.6 | 90.6 |
| Dong *et al.* [6] | 98.8 | 94.1 | **97.8** | 96.9 | 97.6 | 93.3 | 98.0 | 96.3 |
| VoxelPose [40] | **99.3** | 94.1 | 97.6 | 97.0 | 97.6 | 93.8 | **98.8** | **96.7** |
| MvP (Ours) | **99.3** | **95.1** | **97.8** | **97.4** | **98.2** | **94.1** | 97.4 | 96.6 |

## 4.2 Visualization

**3D Pose and Body Mesh Estimation**    We visualize some 3D pose estimations of MvP on the challenging Panoptic dataset in Fig. 4. It can be observed that MvP is robust to large pose deformation (the 1st example) and severe occlusion (the 2nd example), and can achieve geometrically plausible results w.r.t. different viewpoints (the rightmost column). Moreover, MvP is extendable to body mesh recovery and can achieve fairly good reconstruction results (the 2nd and 4th rows). All these results verify both effectiveness and extendability of MvP. Please see supplementary for more examples.

---

[2]We count averaged per-sample inference time in millisecond on Panoptic test set. For all methods, the time is counted on GPU GeForce RTX 2080 Ti and CPU Intel i7-6900K @ 3.20GHz.

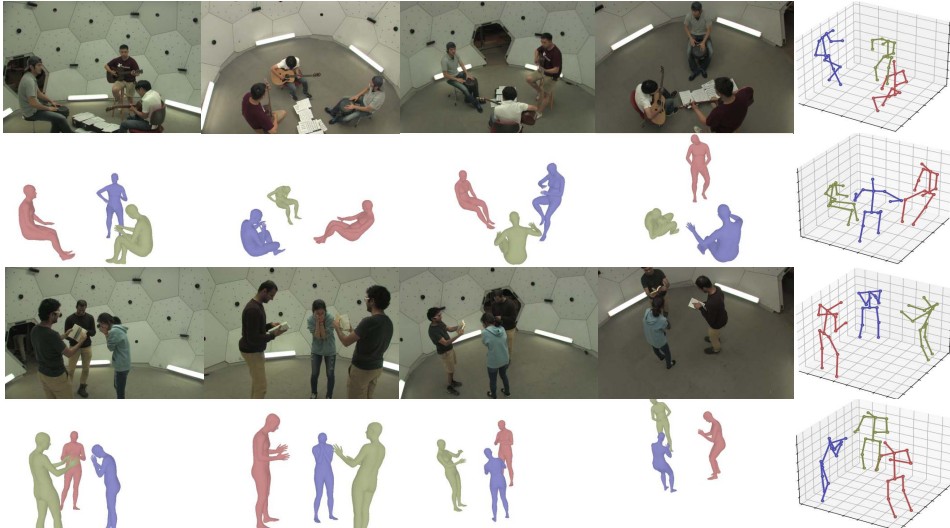

Figure 4: Example 3D pose estimations from Panoptic dataset. The left four columns show the multi-view inputs and the corresponding body mesh estimations. The rightmost column shows the estimated 3D poses from two different viewpoints. Best viewed in color.

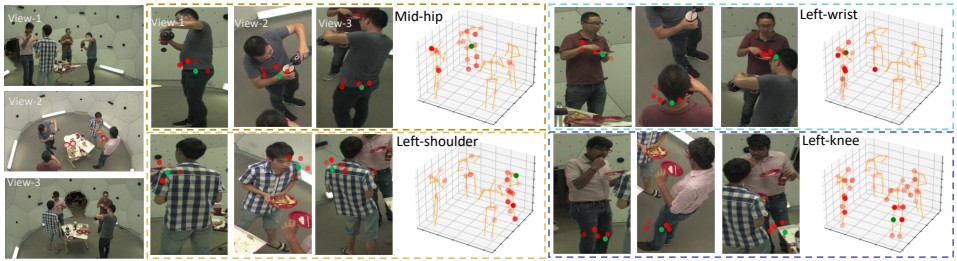

Figure 5: Visualization of projetive attention and self-attention on example skeleton joints. The attention weights are obtained with the 4-th decoder layer of a trained model. *Projective attention* (in the cropped image triplets): the green points denote the projected anchor points in each camera view, and the red points denote the offsetted spatial locations, with brighter color for stronger attention. *Self-attention* (in the 3D skeleton plots): example skeleton joint (green) to all the other skeleton joints (red) in the scene. The color density indicates attention weight. Best viewed in color and $2\times$ zoom.

**Attention Mechanism**    We visualize the projective attention and the self-attention in Fig. 5. Benefiting from the 3D-to-2D projection, the projective attention can accurately locate the skeleton joint in each camera view (the green point) based on the current estimated 3D joint location. We observe it learns to gather adaptive local context information (the red points) with the deformable sampling operation. For instance, when regressing the 3D position of mid-hip (the 1st example), the projective attention selectively attends to informative joints such as the left and right hips as well as thorax, which offers sufficient contextual information for accurate estimation. We also visualize the self-attention, which learns pair-wise interaction between all the skeleton joints in the scene. From the 3D plot in Fig. 5, we can observe a certain skeleton joint mainly attends to other joints of the same person instance (more opaque). It also attends to joints from other person instances, but with less attention (more transparent). This phenomenon is reasonable as the skeleton joints of a human body are strongly correlated to each other, *e.g.*, with certain pose priors and bone length.

## 4.3   Ablation

**Importance of RayConv**    MvP introduces RayConv to encode multi-view geometric information, *i.e.*, camera ray directions into image feature representations. As shown in Table 3a, if removing RayConv, the performance drops significantly—4.8 decrease in $AP_{25}$ and 1.6 increase in MPJPE. This indicates the multi-view geometrical information is important for the model to more precisely

Table 3: Ablations on Panoptic. In (b), *Hier.* denotes the hierarchical query embedding scheme, *Hier.+ad.* means further adding the adaptation strategy. Please see supplement for more ablations.

| RConv | $AP_{25}$ | $AP_{100}$ | MPJPE |
|---|---|---|---|
| w/ | 92.3 | 97.5 | 15.8 |
| w/o | 87.5 | 96.2 | 17.4 |

(a) The effect of *RayConv*. w/o means removing RayConv.

| Query | $AP_{25}$ | $AP_{100}$ | MPJPE |
|---|---|---|---|
| Per-joint | 67.4 | 84.7 | 41.2 |
| Hier. | 82.5 | 93.2 | 19.5 |
| Hier.+ad. | 92.3 | 97.5 | 15.8 |

(b) Different joint query embedding schemes.

| Thr. | $AP_{25}$ | $AP_{100}$ | MPJPE |
|---|---|---|---|
| 0.0 | 93.1 | 98.5 | 16.3 |
| 0.1 | 92.3 | 97.5 | 15.8 |
| 0.2 | 91.1 | 96.2 | 15.5 |
| 0.4 | 89.2 | 93.7 | 15.0 |

(c) Different confidence threshold during evaluation.

| Dec. | $AP_{25}$ | $AP_{100}$ | MPJPE |
|---|---|---|---|
| 2 | 6.3 | 92.5 | 49.6 |
| 3 | 63.4 | 95.6 | 22.8 |
| 4 | 86.8 | 96.8 | 17.5 |
| 5 | 91.8 | 97.6 | 16.2 |
| 6 | 92.3 | 97.5 | 15.8 |
| 7 | 92.0 | 97.5 | 15.9 |

(d) Number of decoder layers.

| Cam. | $AP_{25}$ | $AP_{100}$ | MPJPE |
|---|---|---|---|
| 1 | 4.7 | 61.0 | 93.8 |
| 2 | 37.7 | 93.0 | 34.8 |
| 3 | 71.8 | 95.1 | 21.1 |
| 4 | 84.1 | 96.7 | 19.3 |
| 5 | 92.3 | 97.5 | 15.8 |

(e) Number of camera views.

| $K$ | $AP_{25}$ | $AP_{100}$ | MPJPE |
|---|---|---|---|
| 1 | 88.6 | 96.3 | 18.2 |
| 2 | 89.3 | 97.5 | 17.4 |
| 4 | 92.3 | 97.7 | 15.8 |
| 8 | 84.4 | 91.1 | 20.3 |

(f) Number of deformable points $K$.

localize the skeleton joints in 3D space. Without RayConv, the transformer decoder cannot accurately capture positional information in 3D space, resulting in performance drop.

**Importance of Hierarchical Query Embedding**  As shown in Table 3b, compared with the straightforward and unstructured per-joint query embedding scheme, the proposed hierarchical query embedding boosts the performance sharply—14.1 increase in $AP_{25}$ and 23.4 decrease in MPJPE. Its advantageous performance clearly verifies introducing the person-level queries to collaborate with the joint-level queries can better exploit human body structural information and improve model to better localize the joints. Upon the hierarchical query embedding scheme, adding the query adaptation strategy further improves the performance significantly, reaching $AP_{25}$ of 92.3 and MPJPE of 15.8. This shows the proposed approach effectively adapts the query embeddings to the target scene and such adaptation is indeed beneficial for the generalization of MvP to novel scenes.

**Different Model Designs**  We also examine effects of varying the following designs of the MvP model to gain better understanding on them.

**Confidence Threshold**  During inference, a confidence threshold is used to to filter out the low-confidence and erroneous pose predictions, and obtain the final result. Adopting a higher confidence will select the predictions in a more restrictive way. As shown in Table 3c, a higher confidence threshold brings lower MPJPE as it selects more accurate predictions; but it also filters out some true positive predictions and thus reduces the average precision.

**Number of Decoder Layers**  Decoder layers are used for refining the pose estimation. Stacking more decoder layers thus gives better performance (Table 3d). For instance, the MPJPE is as high as 49.6 when using only two decoder layers, but it is significantly reduced to 22.8 when using three decoder layers. This clearly justifies the progressive refinement strategy of our MvP model is effective. However the benefit of using more decoder layers diminishes when the number of layers is large enough, implying the model has reached the ceiling of its model capacity.

**Number of Camera Views**  Multi-view inputs provide complementary information to each other which is extremely useful when handling some challenging environment factors in 3D pose estimation like occlusions. We vary the number of camera views to examine whether MvP can effectively fuse and leverage multi-view information to continuously improve the pose estimation quality (Table 3e). As expected, with more camera views, the 3D pose estimation accuracy monotonically increases, demonstrating the capacity of MvP in fusing multi-view information.

**Number of Deformable Sampling Points**  Table 3f shows the effect of the number of deformable sampling points $K$ used in the projective attention. With only one deformable point, MvP already achieves a respectable result, *i.e.*, 88.6 in $AP_{25}$ and 17.4 in MPJPE. Using more sampling points

further improves the performance, demonstrating the projective attention is effective at aggregating information from the useful locations. When $K = 4$, the model gives the best result. Further increasing $K$ to 8, the performance starts to drop. It is likely because using too many deformable points introduces redundant information and thus makes the model more difficult to optimize.

## 5 Conclusion

We introduced a direct and efficient model, named Multi-view Pose transformer (MvP), to address the challenging multi-view multi-person 3D human pose estimation problem. Different from existing methods relying on tedious intermediate tasks, MvP substantially simplifies the pipeline into a direct regression one by carefully designing the transformer-alike model architecture with a novel hierarchical joint query embedding scheme and projective attention mechanism. We conducted extensive experiments to verify its superior performance and speed over the well-established baselines.

We empirically found MvP needs sufficient data for model training since it learns the 3D geometry implicitly. In the future, we will study how to enhance the data-efficiency of MvP by leveraging the strategy like self-supervised pre-training or exploring more advanced approaches. Similar to prior works, we also found MvP suffers from performance drop for cross-camera generalization, that is, generalizing on novel camera views. We will explore approaches like disentangling camera parameters and multi-view feature learning to improve this aspect. Besides, we will explore the large-scale applications of MvP and further extend it to other relevant tasks. Thanks to its efficiency, MvP would be scalable to handle very crowded scenes with many persons. Moreover, the framework of MvP is general and thus extensible to other 3D modeling tasks like dense mesh recovery of common objects.

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
