# Direct Multi-view Multi-person 3D Pose Estimation
## (Supplementary Material)

**Tao Wang**[1,2]*, **Jianfeng Zhang**[2]*, **Yujun Cai**[1], **Shuicheng Yan**[1], **Jiashi Feng**[1],
[1]Sea AI Lab [2]National University of Singapore,
twangnh@gmail.com,
zhangjianfeng@u.nus.edu,
{caiyj,yansc,fengjs}@sea.com

## A Implementation Details

We use PyTorch [9] to implement the proposed **M**ulti-**v**iew **P**ose transformer (MvP) model. Our MvP model is trained on 8 Nvidia RTX 2080 Ti GPUs, with a batch size of 1 per GPU and a total batch size of 8. We use the Adam optimizer [7] with an initial learning rate of 1e-4 and decrease the learning rate by a factor of 0.1 at 20 epochs during training. The hyper-parameter $\lambda$ for balancing confidence score and pose regression losses is set to 2.5. We use the image feature representations (256-d) from the de-convolution layer of the 2D pose estimator PoseResNet [11] for multi-view inputs. Additionally, we provide the code of MvP, including the implementation of model architecture, training and inference, in the folder of "./mvp" for better understanding our method.

## B Architecture Details

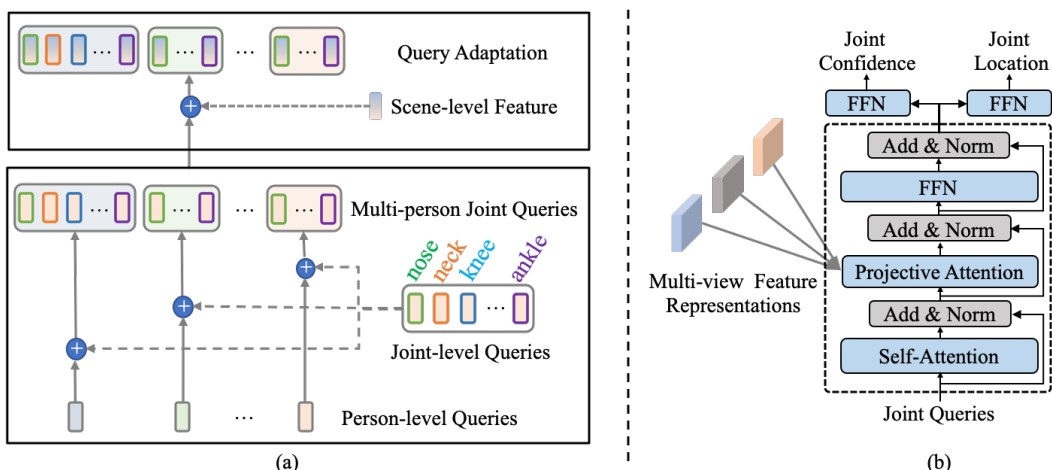

Figure S1: (a) Illustration of the proposed hierarchical query embedding and the input-dependent query adaptation schemes. (b) Architecture of MvP's decoder layer. It consist of a self-attention, a projective attention and a feed-forward network (FFN) with residual connections. Add means addition and Norm means normalization. Best viewed in color.

**Hierarchical Joint Query Embedding**  Fig. S1 (a) illustrates our proposed hierarchical query embedding scheme. As shown in Eqn. (1), each person-level query is added individually to the same

---

*Equal Contribution.

35th Conference on Neural Information Processing Systems (NeurIPS 2021).

set of joint-level queries to obtain the per-person customized joint queries. This scheme shares the joint-level queries across different persons and thus reduces the number of parameters (the joint embeddings) to learn, and helps the model generalize better. The generated per-person joint query embedding is further augmented by adding the scene-level feature extracted from the input images.

**Decoder Layer**    The decoder of MvP transformer consists of multiple decoder layers for regressing 3D joint locations progressively. Fig. S1 (b) demonstrates the detailed architecture of a decoder layer, which contains a self-attention module to perform pair-wise interaction between all the joints from multiple persons in the scene; a projective attention module to selectively gather the complementary multi-view information; and a feed-forward network (FFN) to predict the 3D joint locations and their confidence scores.

## C   More Ablation Studies

**Replacing Camera Ray Directions with 2D Spatial Coordinates**    MvP encodes camera ray directions into the multi-view image feature representations via RayConv. We also compare with the simple positional embedding baseline that uses 2D coordinates as the positional information to embed, similar to the previous transformer-based models for vision tasks [2, 3]. Specifically, we replace the camera ray directions with 2D spatial coordinates of the input images in RayConv. Results are shown in Table S1. We can observe using the 2D coordinates in RayConv results in much worse performance, *i.e.*, 83.3 in $AP_{25}$ and 18.1 in MPJPE. This result demonstrates that using such view-agnostic 2D coordinates information cannot well encode multi-view geometrical information into the model; while using camera ray directions can effectively encode the positional information of each view in 3D space, thus leading to better performance.

Table S1: Results of replacing camera ray directions with 2D coordinates in RayConv.

| Positional Input | $AP_{25}$ | $AP_{100}$ | MPJPE |
|---|---|---|---|
| Camera Ray Directions | 92.3 | 97.5 | 15.8 |
| 2D Spatial Coordinates | 83.3 | 93.0 | 18.1 |

**Replacing Projective Attention with Dense Attention**    We further investigate the effectiveness of the proposed projective attention by comparing it with the dense dot product attention, *i.e.*, conducting attention densely over all spatial locations and camera views for multi-view information gathering. Results are given in Table S2. We observe MvP with the dense attention (MvP-Dense) delivers very poor performance (0.0 $AP_{25}$ and 114.5 MPJPE) since it does not exploit any 3D geometries and thus is difficult to optimize. Moreover, such dense dot product attention incurs significantly higher computation cost than the proposed projective attention—MvP-Dense costs 31 G GPU memory, more than $5\times$ larger than MvP with the projective attention, which only costs 6.1 G GPU memory.

Table S2:  Comparison between the dense attention and the proposed projective attention. MvP-Dense means replacing the projective attention with the dense attention. We report GPU memory cost with a batch size of 1 during training.

| Models | $AP_{25}$ | $AP_{100}$ | MPJPE | GPU Memory[G] |
|---|---|---|---|---|
| MvP-Dense | 0.0 | 16.1 | 114.5 | 31.0 |
| MvP | 92.3 | 97.5 | 15.8 | 6.1 |

## D   More Results

**Quantitative Result**    We also evaluate our MvP model on the most widely used single-person dataset Human3.6M [4] collected in an indoor environment. We follow the standard training and evaluation protocol [8, 5, 10] and use MPJPE as evaluation metric. Our MvP model achieves 18.6 MPJPE which is comparable to state-of-the-art approaches (18.6 *v.s* 17.7 and 19.0) [5, 10].

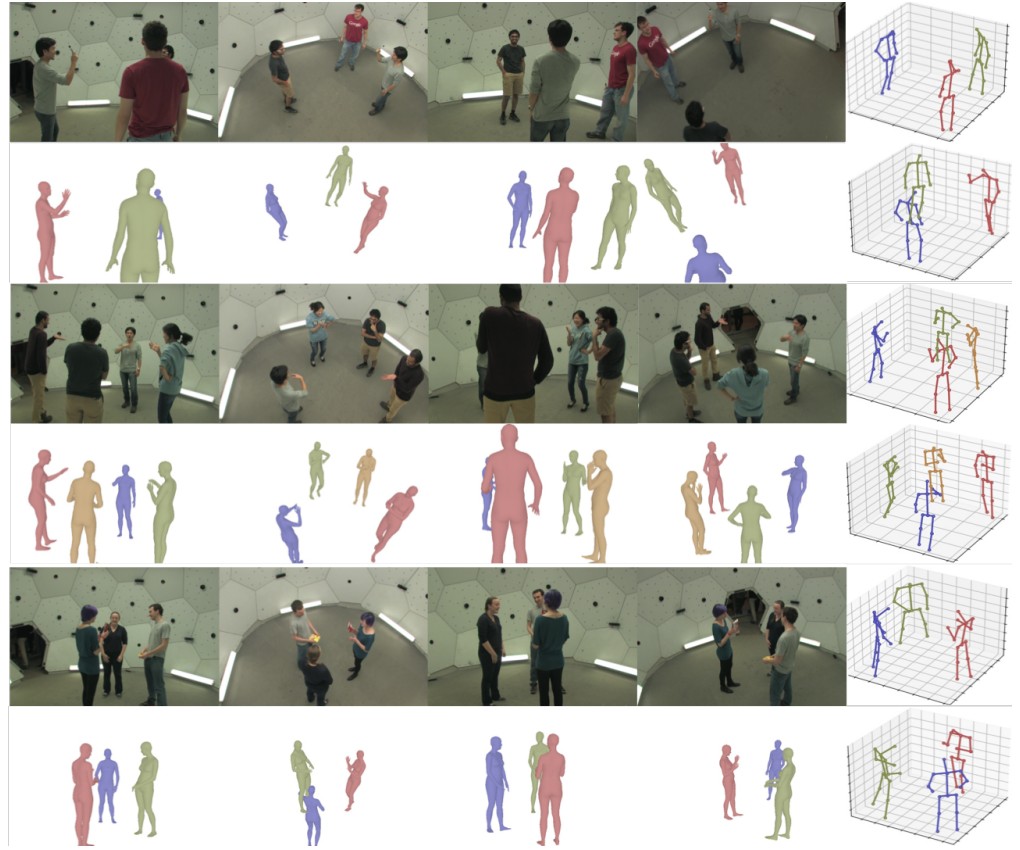

Figure S2: Example 3D pose estimations from Panoptic dataset. The left four columns show the multi-view inputs and the corresponding body mesh estimations from MvP. The rightmost column shows the estimated 3D poses from two different views. Best viewed in color.

**Qualitative Result**    Here we present more qualitative results of MvP on Panoptic [6] (Fig. S2), Shelf and Campus [1] (Fig. S3) datasets. From Fig S2 we can observe that MvP can produce satisfactory 3D pose and body mesh estimations even in case of strong pose deformations (the 1st example) and large occlusion (the 2nd and 3rd examples). Moreover, the performance of MvP is robust even in the challenging crowded scenario, as shown in the 1st example in Fig. S3.