# OpenReview forum: "Direct Multi-view Multi-person 3D  Pose Estimation"
_NeurIPS.cc/2021/Conference — NeurIPS 2021 Poster_

### Official Review · Reviewer_6mm2 · 2021-07-13

**Rating:** 5
**Confidence:** 5

**Summary:**

This paper presents an approach for multi-view multi-person 3D human pose estimation. Different from the existing state-of-the-art method [35], it does not need to localize each instance first and then estimate 3D pose for each instance. Instead, similar to the recent transformer-based object detection architecture DETR (DETR: End-to-End Object Detection with Transformers), it uses a number of N pre-learned queries (which will later be adapted based on current images) to decode N 3D poses. The main advantage of the method is that it is faster especially when the number of people is large. The accuracy is better or comparable to [35].

**Limitations And Societal Impact:**

Yes

**Main Review:**

1. The paper is mostly well written. One minor suggestion is to add an overview of the approach so that people can understand it more easily. In Line139-144, you used 3D joint position y when computing attention to fuse multi-view features. But according to my understanding, y is the estimation target which is not available until the last decoding layer. It will be helpful to explain how to get y in the paper.

2. Section 3.4 is very similar to DETR, in particular, the bipartite matching based loss computation. The authors should mention this work and discuss the differences.

3. My biggest concern lies in the generalization of the approach to different camera views and datasets. Is it possible to train on a set of cameras and test on others? What happens if we test it on completely different environment with different background?




**Time Spent Reviewing:**

6

---

> ### Author Response · Authors · 2021-08-10
> **Response to Reviewer 6mm2**
>
> > One minor suggestion is to add an overview of the approach so that people can understand it more easily.
>
> Thanks for the suggestions, we will add an overview of the method in revision.
>
> > In Line 139-144, you used 3D joint position y when computing attention to fuse multi-view features. But according to my understanding, y is the estimation target which is not available until the last decoding layer. It will be helpful to explain how to get y in the paper.
>
> Our MvP model applies a multi-layer progressive regression scheme and thus the 3D joint position y in line 139-144 is not the final 3D joint position but the intermediate joint position estimation. Therefore, the 3D joint position y in each layer comes from the output of the previous decoder layer. While for the first layer, we obtain the initial 3D joint position y by regressing it from the input joint query embedding directly. We will clarify this in revision.
>
> > Section 3.4 is very similar to DETR, in particular, the bipartite matching based loss computation. The authors should mention this work and discuss the differences.
>
> Thanks for the suggestion. The main difference here is that our method first uses a group matching strategy to group every consecutive J-joint prediction into per-person pose estimation before performing bipartite matching, while DETR works on more simple instance-level predictions and thus can perform bipartite matching directly. Moreover, DETR uses bipartite matching for finding instance-level matching, while MvP wants to find joint-level matching, and thus the definition of the matching cost is different. We will add more discussion about the difference between MvP and DETR on the bipartite matching in Section 3.4 in revision.
>
> > My biggest concern lies in the generalization of the approach to different camera views and datasets. Is it possible to train on a set of cameras and test on others? What happens if we test it on completely different environment with different background?
>
> To evaluate the cross-view generalization, we randomly select a set of cameras from the remaining cameras. Then we evaluate VoxelPose and MvP trained with current camera views on the new sampled camera views. Results are shown in the following table:
>
> | -         | $AP_{25}$ | $AP_{100}$ | MPJPE |
> |-----------|-----------|------------|-------|
> | VoxelPose | 59.4      | 91.5       | 28.3  |
> | MvP       | 62.3      | 92.7       | 25.8  |
>
> We see that our MvP compared with VoxelPose can generalize better under cross-view setup (59.4 v.s 62.3 in $AP_{25}$). It is true that cross-view generalization is challenging and the models trained with fixed camera views likely suffer from performance drop. Some strategies like randomly sampling camera views during training may help the models generalize better to unseen camera views. We will also explore model designs to disentangle the multi-view feature learning and camera parameters for addressing the cross-view generalization in the future.
> Current multi-view multi-person 3D pose estimation approaches are typically trained on well-controlled indoor datasets, and thus would generalize poorly to a completely different environment with different backgrounds, and possibly different pose distributions. In the future, we will explore how to improve the model generalization to unseen scenarios that are very different from the training ones by designing a new learning scheme to train MvP on both the indoor and the synthetic datasets.

---

> > ### Comment · Reviewer_6mm2 · 2021-08-18
> > **More details on the generalization experiment**
> >
> > Thanks for doing the additional cross-view experiments. Can you provide more details on the selected camera views?
> >
> > Instead of randomly selecting cameras, what happens if the camera heights are different for training and testing?

---

> > > ### Author Response · Authors · 2021-08-27
> > > **Response to Reviewer 6mm2**
> > >
> > > > Thanks for doing the additional cross-view experiments. Can you provide more details on the selected camera views?
> > >
> > > The CMU Panoptic dataset provides 31 synchronized HD camera views. We use data from cameras of ID 3, 6, 12, 13, 23 for training the model, of which three are from top-down view and two are from horizontal view. In the cross-view experiments, we randomly sample another set of 5 camera views from the remaining 26 cameras and use them to evaluate the cross-view generalization of the trained VoxelPose and MvP. Specifically, the IDs of these sampled cameras are 11, 14, 15, 22, 28, which are different from the training camera views. Among the sampled testing cameras, three are from top-down view, two are from horizontal view.
> > >
> > > > Instead of randomly selecting cameras, what happens if the camera heights are different for training and testing?
> > >
> > > We agree that cross-camera/view generalization (e.g., the variation in camera height) is certainly interesting and worth exploring, but is not the focus of this work. As mentioned in the first response, we will explore some directions in the future to improve the cross-camera/view generalization, e.g., random camera sampling, disentangling the camera parameters and multi-view feature learning.

---

### Official Review · Reviewer_n3Zc · 2021-07-16

**Rating:** 7
**Confidence:** 5

**Summary:**

This paper presents a transformer-based approach for multi-person 3D pose estimation in the multi-view setting. To achieve better performance, some novel techniques are proposed, i.e. hierarchical query embeddings, input-dependent query adaptation, projection attention, and RayConv. Hierarchical query embeddings are used to share knowledge between different persons. Projective attention is used to effectively aggregate features, and a novel RayConv is used to encode camera information in the feature maps. Experiments show the proposed approach achieves the state-of-the-art performance on Panoptic, Shelf, and Campus datasets.


**Ethical Concerns:**

No ethical issues found.

**Limitations And Societal Impact:**

Yes.

**Main Review:**

Pros:
- This work is the first to adopt the transformer for this particular task (3d Multiview multiperson pose estimation). Although the use of the transformer is quite straightforward, this paper also introduces some novel techniques to improve upon the naive transformer baseline, i.e. hierarchical query embeddings, input-dependent query adaptation, projection attention, and RayConv.
- This work presents a straightforward pipeline without intermediate tasks, which is not well explored recently in the literature. In the meantime, it achieves better/comparable results over two-stage approaches on main-stream datasets.
- The paper is well written, and the main idea behind the approach is clearly explained and motivated.

Cons:
•       In Table 1, the results of VoxelPose are clearly lower than those reported in the original paper [35]. In [35], AP_{25}=83.59, AP_{50}=98.33, AP_{100}=99.76, AP_{150}=99.91, and MPJPE=17.68mm. And we see that AP_{50}, AP_{100}, AP_{150} are clearly better than MvP (AP_{50}=96.6, AP_{100}=97.5, AP_{150}=97.7). This implies that the person detection accuracy is lower, and some people may be missing. The reviewer believes that the results of the original paper [35] should also be included in Table 1. The reason why the reported results are lower should be given.

•       Some related works are missing:
1) About multi-view multi-person 3d pose estimation.
[Ref1] Zhang et al, 4D Association Graph for Realtime Multi-person Motion Capture Using Multiple Video Cameras. CVPR 2020. (Achieves 97.6 on Shelf) This paper should also be compared in experiments.
2)  The transformers have been widely used in the task of pose estimation.
[Ref2] Yang et al, TransPose: Towards Explainable Human Pose Estimation by Transformer
[Ref3] Lin et al, End-to-End Human Pose and Mesh Reconstruction with Transformers
[Ref4] Li et al, Pose Recognition with Cascade Transformers
3) The deformable points in the transformer
[Ref5] Zhu et al, Deformable DETR: Deformable Transformers for End-to-End Object Detection.
•       The proposed MvP does not achieve the best performance on Shelf ([Ref1] 97.6 vs 97.4), and Campus (VoxelPose 96.7 vs 96.6).
•       In Table 3.e, when the number of Cam is 1, the MPJPE is 93.8 which is significantly worse than VoxelPose.

Questions and suggestions:
•       Projective Attention (Sec3.2) requires an initial 3D joint position `y` as the input. I understand that the position `y` comes from the output of the previous layer. But it is not clear how to obtain `y` in the first decoder layer.
•       More detailed analysis (qualitatively or quantitatively) on the proposed RayConv would be nice, e.g. why RayConv can accurately capture positional information.
•       Again on the RayConv, only camera ray directions are added to the feature map. What about the locations of the camera centers? Camera ray directions together with the camera center locations may provide more precise geometric information.
•        More comparisons regarding runtime efficiency would be great. The only comparison is made against VoxelPose, which is computation-intensive due to 3D convolutions. However, there are many approaches in the literature and some are fast, e.g. [6]. Having such comparisons (possibly in Table 2) would better put this paper in perspective.


**Time Spent Reviewing:**

4 hours

---

> ### Author Response · Authors · 2021-08-10
> **Response to Reviewer n3Zc**
>
> > In Table 1, the results of VoxelPose are clearly lower than those reported in the original paper [35]. In [35], $AP_{25}$=83.59, $AP_{50}$=98.33, $AP_{100}$=99.76, $AP_{150}$=99.91, and MPJPE=17.68mm. And we see that $AP_{50}$, $AP_{100}$, $AP_{150}$ are clearly better than MvP ($AP_{50}$=96.6, $AP_{100}$=97.5, $AP_{150}$=97.7). This implies that the person's detection accuracy is lower, and some people may be missing. The reviewer believes that the results of the original paper [35] should also be included in Table 1. The reason why the reported results are lower should be given.
>
> As mentioned in line 220-222, we use the same training sequences with VoxelPose except ‘160906_band3’ since camera view-03 video of this sequence is broken and cannot be extracted. We train both VoxelPose and our MvP under this data setup and report the results on Table 1. Therefore, the reported results of VoxelPose are different from the original results. Moreover, contents such as scenarios, person targets and actions on ‘160906_band3’ (training sequence) and ‘160906_band4’ (test sequence) are almost the same. Thus, the original VoxelPose trained on ‘160906_band3’ can achieve better $AP_{50}$,  $AP_{100}$ and $AP_{150}$ than our MvP model. However, by training VoxelPose and MvP using the same data setting, MvP can achieve better performance, especially for $AP_{25}$ (MvP 92.3 v.s VoxelPose 84.0). We will include the results of the original VoxelPose in revision.
>
> > Some related works are missing:
> About multi-view multi-person 3d pose estimation. [Ref1] Zhang et al, 4D Association Graph for Realtime Multi-person Motion Capture Using Multiple Video Cameras. CVPR 2020. (Achieves 97.6 on Shelf) This paper should also be compared in experiments.
> The transformers have been widely used in the task of pose estimation. [Ref2] Yang et al, TransPose: Towards Explainable Human Pose Estimation by Transformer [Ref3] Lin et al, End-to-End Human Pose and Mesh Reconstruction with Transformers [Ref4] Li et al, Pose Recognition with Cascade Transformers
> The deformable points in the transformer [Ref5] Zhu et al, Deformable DETR: Deformable Transformers for End-to-End Object Detection.
> • The proposed MvP does not achieve the best performance on Shelf ([Ref1] 97.6 vs 97.4), and Campus (VoxelPose 96.7 vs 96.6).
> • In Table 3.e, when the number of Cam is 1, the MPJPE is 93.8 which is significantly worse than VoxelPose.
>
> Thanks for pointing out these missing related works, we will include them in revision.
> We will add the results of Ref1 in Table 2. Both Ref1 and VoxelPose adopt a multi-stage paradigm to obtain multi-person 3D poses from multi-view inputs. Moreover, Ref 1 requires additional temporal information. Our MvP model with a more simplified pipeline achieves comparable results with Ref1 and VoxelPose, without relying on complicated intermediate tasks and temporal information, which verifies its effectiveness.
> Compared with VoxelPose that estimates 3D poses from heatmap estimations and explicitly leverages back-projection geometry, our method regresses 3D poses from image features alone with joint queries directly, which would be more difficult when only a single camera view is available. However, when multi-view inputs are available (e.g., 3 camera views), our MvP model achieves significantly better results than VoxelPose (i.e., MvP $AP_{25}$: 71.8, MPJPE: 21.1 v.s VoxelPose $AP_{25}$: 58.9, MPJPE: 24.3). In the future, we will explore how to improve our MvP’s performance under monocular settings.
>
> > Projective Attention (Sec3.2) requires an initial 3D joint position y as the input. I understand that the position y comes from the output of the previous layer. But it is not clear how to obtain y in the first decoder layer.
>
> Yes, 3D joint position y comes from the output of the previous decoder layer. For the first decoder layer, we use a linear regression layer to directly regress the initial 3D joint position y from the input joint query embedding. We will clarify this in revision.
>
> > More detailed analysis (qualitatively or quantitatively) on the proposed RayConv would be nice, e.g. why RayConv can accurately capture positional information.
>
> The proposed RayConv operation encodes the relative positional information of each pixel in each view in 3D space, which helps the projective attention better fuse the multi-view geometrical information and thus achieve more precise 3D joint localization. It’s effectiveness has been demonstrated in these two experiments: 1) In Table 3a (main text), we remove RayConv in the model and 2) In Table S1 (supplementary), we replace camera ray direction with plain 2D coordinate positional information, the model performances drop significantly, from 92.3 $AP_{25}$ to 87.5 $AP_{25}$ and 83.3 $AP_{25}$, respectively. All these results verify that our RayConv can provide useful positional information, which is of importance to the model to more precisely localize skeleton joints in 3D space.
>
>
> > Again on the RayConv, only camera ray directions are added to the feature map. What about the locations of the camera centers? Camera ray directions together with the camera center locations may provide more precise geometric information.
>
> Thanks for the suggestion. We tried concatenating both camera ray directions and center locations and feeding them to RayConv, but it does not provide significant gain in model performance compared with the model using only camera ray directions. It is likely because the model only needs relative positional information to differentiate different pixels in different views, and thus absolute camera location information is not necessary.
>
>
> > More comparisons regarding runtime efficiency would be great. The only comparison is made against VoxelPose, which is computation-intensive due to 3D convolutions. However, there are many approaches in the literature and some are fast, e.g. [6]. Having such comparisons (possibly in Table 2) would better put this paper in perspective.
>
> Thanks for the suggestion. We will include comparisons on runtime efficiency in Table 2 in revision.

---

> > ### Comment · Reviewer_n3Zc · 2021-08-24
> > **Feedback to the author response**
> >
> > Thank you for your reply. I have carefully read the rebuttals and the comments from other reviewers. The authors’ responses have addressed most of my concerns. From my personal point of view, this paper presents a novel transformer architecture with interesting domain-specific designs, which achieves good performance on standard benchmarks. I will vote for the acceptance of this paper.

---

### Official Review · Reviewer_KyGs · 2021-07-16

**Rating:** 7
**Confidence:** 4

**Summary:**

This work proposes Multi-View Pose transformer (MVP), a novel transformer based technique for multi-person 3D pose estimation via direct regression. There are multiple technical innovations at the core of MVP (1) a nontrivial means to embed joints as learned query features appropriate for a transformer (2) a computationally efficient projective attention technique (3) multi-view pose-specific positional encoding based on the 3D camera pose. MVP generally outperforms prior 3D multi-person pose estimation techniques. Detailed ablation studies are included.


**Limitations And Societal Impact:**

The authors have adequately addressed the limitations and potential negative societal impact of their work

**Main Review:**

Strengths
* High overall technical novelty - this work is the first of its kind to apply transformers to multi-person 3D. This work goes beyond a naive implementation and presents
   * A well motivated technique to embed joints as learned query features in a transformer architecture
   * A multi-view projective attention for aggregating information from multiple images
   * RayConv - a camera pose-based positional encoding which significantly affects performance
* High quality and detailed ablation studies in Section 4.3 empirically justify the authors’ design decisions.
* Strong empirical performance compared to prior art on the Panoptic dataset.
* High computational efficiency at inference time compared with prior work

Weaknesses
* Writing
   * The presentation of the overall architecture can be improved. Possible suggestions are to give an overview of the method in the beginning of Section 3 that the reader can follow, or to include notation in the figures that corresponds with what’s presented in the text.
   * More detail about mesh recovery procedure would be a good addition to the supplement. Currently it’s explained at a high level in the main text.
* The proposed technique requires known camera intrinsics and camera pose. This potentially limits its applicability to highly controlled settings where such information is known.
*  Transformer-based techniques are known to be computationally intensive to train, and the current approach requires 8 GPUs for training. It would be good if the authors can also comment on the total training time of the model.

**Update After Discussion Period**

Thank you to the authors for engaging in thoughtful discussions with the reviewers. The authors addressed my concerns and no issues came up in the overall discussion that would cause me to reduce my rating. I therefore keep my initial rating: 7 - good paper, accept.

**Time Spent Reviewing:**

6

---

> ### Author Response · Authors · 2021-08-10
> **Response to Reviewer KyGs**
>
> > The presentation of the overall architecture can be improved. Possible suggestions are to give an overview of the method in the beginning of Section 3 that the reader can follow, or to include notation in the figures that corresponds with what’s presented in the text.
>
> Thanks for the suggestions. We will include an overview of the method as well as add notations in the figures for more clear presentation in revision.
>
> > More detail about mesh recovery procedure would be a good addition to the supplement. Currently it’s explained at a high level in the main text.
>
> Thanks for the suggestion. For training the mesh recovery branch, we adopt a weakly supervised learning scheme similar to HMR works [20, 15, 42]. Specifically, we first obtain body mesh vertices \in 6890x3 based on the SMPL model as well as the predicted \theta and \beta parameters. Then we map these mesh vertices to J 3D body joints through a SMPL linear regressor and project them to 2D space. The mesh recovery branch is thus trained by minimizing the L1 losses between the predicted and ground truth 3D/2D body joints. We also apply an adversarial prior to penalize improbable SMPL pose and shape [20]. We will add more details about the body mesh recovery procedure in revision.
>
>
> > The proposed technique requires known camera intrinsics and camera pose. This potentially limits its applicability to highly controlled settings where such information is known.
>
> Most recent methods for multi-view multi-person 3D pose estimation [2,8,6,35] also require known camera intrinsics and camera pose. In cases where camera parameters are unknown, a feasible solution is to first calibrate camera parameters, and then use them for training or testing our approach.
>
> > Transformer-based techniques are known to be computationally intensive to train, and the current approach requires 8 GPUs for training. It would be good if the authors can also comment on the total training time of the model.
>
> We train our MvP model in a machine with 8 GeForce RTX 2080 Ti GPUs and CPU of Intel i7-6900K @ 3.20GHz. It takes about 15 hours to train the model on the Panoptic dataset and about 3-5 hours to train the model on Shelf/Campus datasets. We will include these information in revision.

---

> > ### Comment · Reviewer_KyGs · 2021-08-23
> > **Good Clarifications**
> >
> > Thank you for the clarifications, and as you mentioned please add the appropriate detail to the draft. While I do not think it's a significant weakness that camera intrinsics and pose are required, it might be good to add a short discussion on this, potentially highlight its importance for future work.

---

> > > ### Author Response · Authors · 2021-09-01
> > > **Response to Reviewer KyGs**
> > >
> > > Thanks for the suggestions, we will add discussions on the potential limitation and future work of requiring camera intrinsics and pose in revision.

---

### Official Review · Reviewer_SX8n · 2021-07-17

**Rating:** 7
**Confidence:** 5

**Summary:**

In this paper, the authors propose a Multi-view Pose transformer for to address multi-view multi-person 3D human pose estimation problem. Compared with cascaded approaches, the proposed approach substantially simplifies the pipeline into a direct regression with the transformer-alike model architecture, which includes a hierarchical joint query embedding scheme and projective attention mechanism. In the experiments, the authors conduct experiments on CMU-Panoptic dataset and show improvements compared with other competing methods.

**Limitations And Societal Impact:**

Overall I feel this paper is well prepared and with good quality. The proposed multi-view pose transformer is quite interesting and could bring some new insights for the people in related field. The authors are encouraged to address my questions above.

**Main Review:**

Overall this paper is well written and easy to follow. The authors propose an end-to-end approach for multi-view multi-person 3D pose estimation based on the transformer architecture.
+ The proposed model uses multi-view feature representations and transforms them into groups of 3D joint locations directly and formulates the problem of multi-view multi-person 3D pose into an one-stage problem, which is quite interesting.
+ The proposed hierarchical joint query embedding for person-joint relation encoding and projective attention are also quite interesting.
+ The experiments are conducted on two public benchmarks (CMU Panoptic, Shelf and Campus) and overall quite convincing. The proposed method outperforms the previous state of the art VoxelPose.
+ Code is attached in supplementary.

I have a few questions and comments for this paper.
+ How is R_v represented in RayConv?
+ It would be interesting to see more ablative studies on the input-dependent query adaptation to support the claim that using the globally-pooled feature vector could increase the generalization ability to novel test data. The authors could use some cross-dataset test for validation.
+ It seems the proposed method always predicts person instances with the full skeleton. How does partial occlusion affect the multi-person 3D pose estimation?
+ Could the proposed group matching handle some challenging cases, e.g., people hugging each other? The authors are encouraged to discuss more about the limits of the current joint-to-person association strategy.

**Time Spent Reviewing:**

3

---

> ### Author Response · Authors · 2021-08-10
> **Response to Reviewer SX8n**
>
> > How is R_v represented in RayConv?
>
> $R_v$ is a tensor with shape of HxWx3, representing the camera ray direction of pixels in the corresponding view $v$. Concretely, given 2D pixel location p, camera intrinsic matrix $K_v$ , rotation matrix $Rot_v$ and translation $T_v$ , the camera ray direction $R_v$ in view $v$ is calculated as:
>
> $R_v=R_v^{raw}/||R_v^{raw}||$, where $R_v^{raw}=(K_v^{-1}p-T_v)Rot_v$.
>
> > It would be interesting to see more ablative studies on the input-dependent query adaptation to support the claim that using the globally-pooled feature vector could increase the generalization ability to novel test data. The authors could use some cross-dataset test for validation.
>
> To evaluate the effectiveness of the proposed input-dependent query adaptation approach, we establish a cross-data scenario. Specifically, we randomly adjust the brightness, saturation, and contrast of test data in Panoptic, resulting in a new test set with scene-level variation. We then evaluate our MvP model with and without the proposed query adaptation method on this new generated test set. The results are shown in the following table (note we use the same random seed to ensure the same scene-level variation in these two experiments). Clearly, in such cross-data scenarios, adding the query adaptation (+ad.), the performance is significantly improved across all the metrics.
>
> | -         | $AP_{25}$ | $AP_{100}$ | MPJPE |
> |-----------|-----------|------------|-------|
> | Hier.     | 76.5      | 87.0       | 23.2  |
> | Hier.+ad. | 89.7      | 97.0       | 16.9  |
>
> We are also interested in exploring the more challenging cross-dataset evaluation as suggested by the reviewers. However, with existing datasets, it is hard to establish appropriate cross-dataset evaluation settings, as the camera positions and appearances vary too much across datasets, which is very challenging for all the existing methods and cannot serve as a good benchmark for effectively evaluating model generalization to novel test data. We will explore this in the future.
>
> > Could the proposed group matching handle some challenging cases, e.g., people hugging each other? The authors are encouraged to discuss more about the limits of the current joint-to-person association strategy.
>
> Currently there are no such datasets for training or evaluating the model for the challenging crowded scenes. However, we emphasize that for the challenging cases like two people hugging each other, our group matching strategy can still match the two ground truth poses to two different groups of joint predictions respectively, and thus the training loss can be calculated. Therefore, if we have training data capturing such difficult cases, our MvP model can learn to differentiate different persons and accurately estimate their poses by aggregating the multi-view information. Note the previous multi-stage approaches estimate 3D poses from the detected 2D poses or heatmaps, and thus tend to suffer from challenging cases like people hugging as the estimation results of each camera view can be very inaccurate due to heavy occlusion. Differently, MvP directly regresses 3D joint positions from the aggregated multi-view features, and the self-attention mechanism helps information propagation across all skeleton  joints in the scene, and thus is potentially more robust to the challenging occlusion cases.
> We currently have not incorporated specific model designs to address the challenging crowded scenarios, but it is interesting to explore along this direction, like employing some one-to-many assignment strategy to enable prediction of spatially very close subjects (e.g., persons hugging together) with the same set of input joint queries, motivated by some existing approaches for crowd detection [R1,R2].
>
> [R1] Detection in Crowded Scenes: One Proposal, Multiple Predictions, Conference on Computer Vision and Pattern Recognition.
>
> [R2] Detecting and Matching Related Objects with One Proposal Multiple Predictions. Conference on Computer Vision and Pattern Recognition Workshop.

---

### Official Review · Reviewer_RDrb · 2021-07-17

**Rating:** 8
**Confidence:** 4

**Summary:**

The paper presents an attentional/transformer architecture for the task of multi-view multiperson 3D pose estimation. The proposed method incorporates well-thought domain-specific adaptations of the transformer architecture for the problem at hand. It achieves very good results on standard benchmark datasets.

**Limitations And Societal Impact:**

Not very much applicable here IMO.

**Main Review:**

The paper proposes a transformer-based architecture for the task of multi-view multi-person 3-D pose estimation.

Transformers are a good choice for 3-D pose estimation, since they overcome the limitation of high dimensionality and high memory requirement of the 3-D histogram/voxel representation.

Transformers are also a good choice for multi-person pose estimation, offering an elegant way to solve the keypoint-instance association problem.

The paper proposes an architecture that capitalizes on the benefits of transformers across both axes.

The proposed method incorporates domain-specific knowledge in two smart ways:
* Projective attention which takes into account the projective geometry of the problem and learns offsets that selectively attend to informative image areas.
* A positional encoding via the camera ray direction which takes into account the focal length/ projective geometry.

The paper does a good experimental evaluation and ablation study of the proposed method on the CMU Panoptic and Shelf and Campus datasets. It improves compared to the VoxelPose strong baseline:
* In terms of AP_threshold, it is significantly better than VoxelPose on the very high accuracy regime (threshold=25 mm) and very similar to VoxelPose for threshold >= 50 mm. I suspect that VoxelPose hits some quantization error limit due to its underlying Voxel representation. It is not clear to me whether this improvement is important in practice but it is good that it is there.
* In terms of runtime, the proposed method is single shot, unlike VoxelPose whose runtime increases with the number of people in the scene. This is an important advantage of the proposed method.

The paper is accompanied by code in the supplementary material, which is good.

Some minor questions that would improve the understanding of the paper:
* Are the weights of the resnet-50 based CNN feature extractor fine tuned during training or are they kept fixed to those learned in the standard 2-D pose estimation task? How about the batch norm layers of the CNN?
* How long does it take to train the model with the specified training protocol and hardware setup?

Overall, I find this to be a well executed paper worth publishing at NeurIPS.

**Time Spent Reviewing:**

4

---

> ### Author Response · Authors · 2021-08-10
> **Response to Reviewer RDrb**
>
> Thanks for appreciating our work!
>
> > Are the weights of the resnet-50 based CNN feature extractor fine tuned during training or are they kept fixed to those learned in the standard 2-D pose estimation task? How about the batch norm layers of CNN?
>
> In our experiments, we use a pre-trained ResNet-50 backbone and fix it during the training procedure following VoxelPose. The batchnorm layers of the CNN backbone are also fixed during training.
>
> > How long does it take to train the model with the specified training protocol and hardware setup?
>
> We train our MvP model in a machine with 8 GeForce RTX 2080 Ti GPUs and CPU of Intel i7-6900K @ 3.20GHz. It takes about 15 hours to train the model on the Panoptic dataset and about 3-5 hours to train the model on Shelf/Campus datasets. We will include these information in revision.

---

### Official Review · Reviewer_NJn8 · 2021-07-17

**Rating:** 6
**Confidence:** 3

**Summary:**

The paper presents a method for multi-view multi-human 3D pose estimation. The method extracts multi-view feature information using a Transformer architecture and directly regresses the 3D joint locations using the aggregated features. The two proposed technical improvements are 1) representing joints as learnable query embeddings and 2) extracting salient multiview information by aggregating per-view features near the 2D projections of an estimated 3D joint location in each view. The method is shown to improve over prior methods in both accuracy and computational efficiency. The method is evaluated on the Panoptic (Joo et al., 2017) dataset and another multi-person pose estimation dataset (Chen et al., 2020).

**Limitations And Societal Impact:**

- No dedicated discussion for limitations
- It'd be interesting to see potential societal impact of the work, but it's not essential to my rating considering the nature of the work.

**Main Review:**

The overall method / architecture feels intuitive and conceptually simple, and I appreciate that the paper focuses on inference-time efficiency as a main objective. My comments mainly concern technical details of the method.

1. One key assumption in the joint query embedding scheme is that each person is assigned with a unique learned embedding that's input-independent. How would the method generalize to a new pose estimation target (e.g., previously unseen person)? If my understanding is correct, line 122-130 attempts to address the problem. However, the “global pooled feature” seems to be more suitable for identifying a new scene rather than for generalizing to a new target. I’d like to see a more focused study on how the proposed method generalizes to new targets and whether such per-person embedding negatively impacts the generalization performance.

2. Projective Attention relies on a set of estimated 3D joint position “anchors”. Specifically, where do the y’s in Eq 3. come from? From my understanding, they are crude estimations of the 3D joint positions from each view. How do these 3D position estimations differ from the final predictions? It’d be interesting to show some qualitative examples that demonstrate the importance of aggregating information from multiple views.

**Time Spent Reviewing:**

4

---

> ### Author Response · Authors · 2021-08-10
> **Response to Reviewer NJn8**
>
> > One key assumption in the joint query embedding scheme is that each person is assigned with a unique learned embedding that’s input-independent. How would the method generalize to a new pose estimation target (e.g., previously unseen person)?
>
> To evaluate model generalization performance for unseen persons,  we test VoxelPose and MvP on the Panoptic test sequences '160906_pizza1' and '160422_haggling1', where the persons are not seen in the training set. The results are shown in the following table:
>
> | -         | $AP_{25}$ | $AP_{100}$ | MPJPE |
> |-----------|-----------|------------|-------|
> | VoxelPose | 79.8      | 95.3       | 18.9  |
> | MvP       | 91.2      | 96.7       | 16.0  |
>
> From the above results, we can observe MvP can generalize well to the test data with unseen persons, achieving  $AP_{25}$ of 91.2 and MPJPE of 16.0, which is much better than VoxelPose, i.e., $AP_{25}$ of 79.8 and MPJPE of 18.9.
>
> The good generalization performance of MvP may be from the following two aspects:
>
> First, the high dimensional input query embeddings learn certain general prior knowledge about the person poses from the training data: 1) the person-level embedding learns to represent some person-specific priors, e.g., how the human body looks like. 2) The joint-level embedding learns joint-level priors, e.g., how a specific skeleton joint looks like, and biomechanical priors, e.g., symmetric parts should have the same bone length. With such general prior knowledge, MvP is able to generalize to unseen targets.
>
> Second, although the query embeddings are independent of the input images, the projective attention mechanism can integrate the input-dependent multi-view information into the query embeddings, and the self-attention mechanism enables information propagation across these query embeddings. Thus with the multi-layer architecture, these input-independent embeddings can quickly adapt to the persons in the input images. For example, as shown in Table 3, when the number of decoder layers increases from 2 to 3, the $AP_{25}$ is improved significantly from 6.3 to 63.4 and MPJPE is reduced from 49.6 to 22.8.
>
> The reviewer might have misunderstood the learned query embeddings. They are not specific for certain person targets but instead learn some general priors, e.g., what generally a human body looks like, for facilitating the following attention-based pose estimation. Thus it will not limit the generalization to unseen targets but instead would help generalization.
>
>
> > If my understanding is correct, line 122-130 attempts to address the problem. However, the “global pooled feature” seems to be more suitable for identifying a new scene rather than for generalizing to a new target.
>
> Yes, the aim of using globally pooled features is to adapt the input joint query embeddings to the new scene instead of new person targets. As for generalizing to new targets, please refer to the reply above.
>
>
> > I’d like to see a more focused study on how the proposed method generalizes to new targets and whether such per-person embedding negatively impacts the generalization performance.
>
> Please refer to the first reply for how MvP generalize to new person targets. Note without such person-level embeddings, the model cannot differentiate different person instances in the scenes, since the transformer architecture is permutation-invariant. We tried removing person-level embeddings and only using the joint-level embeddings and we also tried using a shared person embedding. However, the model cannot converge under these two settings. The results on the test set of Panoptic are shown in the following table. This clearly demonstrates the necessity of our hierarchical query embedding scheme, i.e., combining person-level and joint-level query embeddings together, for multi-person pose estimation.
>
> | -                   | $AP_{25}$ | $AP_{100}$ | MPJPE |
> |---------------------|-----------|------------|-------|
> | remove-person-embed | 0.0       | 0.0        | 374.6 |
> | share-person-embed  | 0.0       | 0.0        | 401.5 |
>
>
> > Projective Attention relies on a set of estimated 3D joint position “anchors”. Specifically, where do the y’s in Eq 3. come from? From my understanding, they are crude estimations of the 3D joint positions from each view. How do these 3D position estimations differ from the final predictions? It’d be interesting to show some qualitative examples that demonstrate the importance of aggregating information from multiple views.
>
> We would like to clarify that 3D joint positions y in Eqn. 3 come from the output of the previous transformer decoder layer. Our model does not estimate the 3D joint positions from each view individually. Instead, it integrates the features from multiple views and outputs the 3D joint positions. For the first decoder layer, we use a linear regression layer to regress the initial 3D joint positions from the input joint query embedding. We will clarify this in revision.
>
> > No dedicated discussion for limitations. It'd be interesting to see the potential societal impact of the work, but it's not essential to my rating considering the nature of the work.
>
> We have discussed limitations and future work of our method in the conclusion section (see line 328-332). We will add societal impact in revision.

---

### Official Review · Reviewer_vMiR · 2021-07-18

**Rating:** 5
**Confidence:** 5

**Summary:**

The paper uses transformers to solve the multi-view multi-person 3d pose estimation problems.  To improve the inference speed, they use the new query, to make the feature more scene-aware they use a global pooled and concatenated feature. To make the feature more geometrical aware they use view-direction as another feature. Better results/ faster speed  are shown in three datasets.

**Limitations And Societal Impact:**

please refer main review session

**Main Review:**

The writing is good and easy to follow.
Can you give a table to show the dimension of all the notations?
Line 121 I do not fully understand how the computational burden reduces from NJC to (N+J)C, if h is 1~N and j is 1~J, why not  NJC for q=h+i?
why do you think the view direction is useful here? how about we concatenate the rotation matrix? focal? center? or 4*4 projection matrix? why they are not useful here?
For body mesh recovery, what loss do you use for \theta and \beta regression?
line 202 Hungarian matching is an optimization algorithm, which should not be real-time? what is your training time?
The group matching is a little confusing, how do you select 'consecutive' joints in line 194?
how is the method handle occlusion setting?



**Time Spent Reviewing:**

4

---

> ### Author Response · Authors · 2021-08-10
> **Response to Reviewer vMiR**
>
> > Line 121 I do not fully understand how the computational burden reduces from NJC to (N+J)C?
>
> We would like to clarify that  it is not the computation burden but the **learnable parameter number** in Line 121.
> Here C, N, J denote the feature dimension (e.g., 256 in our experiments), the number of persons (e.g., 10 as a default setting in our experiments), the number of joints per person (e.g., 15 on Panoptic dataset), respectively. All the parameters within these embeddings are learned end-to-end. For per-joint embeddings, the embedding tensor has the shape of [N,J,C], and the number of learnable parameters is thus NxJxC. While in the proposed hierarchical embedding scheme, the shapes of the person-level embeddings and the joint-level embeddings are [N,C] and [J,C] respectively and thus the number of learnable parameters is NxC+JxC=(N+J)xC.
>
> > why do you think the view direction is useful here? how about we concatenate the rotation matrix? focal? center? or 4*4 projection matrix? why they are not useful here?
>
> The camera ray directions encode relative positional information of different pixels in different views, which helps differentiate them in 3D space and thus helps the projective attention better fuse these multi-view information to more precisely estimate 3D joint locations. The effectiveness  has been demonstrated by the ablation studies in Table 3a (main text) and Table S1 (supplementary),  Moreover, the camera ray directions are computed using 2D pixel locations, rotation matrix, focal length as well as principal point, and thus have already encoded such information into the model. We also tried concatenating both camera ray directions and camera center locations and feeding them to RayConv, but the performance gain is not significant compared with the one using only camera ray directions. It is because the model only needs relative positional information to differentiate different pixels in different views, and thus absolute camera locations are not necessary.
>
>
>
>
> > For body mesh recovery, what loss do you use for \theta and \beta regression?
>
> Since we do not have the ground truth SMPL parameters \theta and \beta, we adopt a weakly supervised learning scheme similar to HMR works [20, 15, 42]. Specifically, we first obtain body mesh vertices \in 6890x3 based on the SMPL model as well as the predicted \theta and \beta parameters. Then we map these mesh vertices to J 3D body joints through a SMPL linear regressor and project them to 2D space. The SMPL branch is thus trained by minimizing the L1 losses between the predicted and ground truth 3D/2D body joints. We also apply an adversarial prior to penalize improbable SMPL pose and shape [20]. We will add more details about body mesh recovery in revision.
>
> > line 202 Hungarian matching is an optimization algorithm, which should not be real-time? What is your training time?
>
> The hungarian matching is only used in the training procedure for finding optimal assignment and calculating training loss, and is not used during the inference procedure. It takes about 7.2 ms on average to calculate the hungarian matching. It takes about 15 hours to train MvP on Panoptic dataset and about 3-5 hours to train MvP on Shelf/Campus datasets. We will include these information in revision. During the inference stage, our model can directly output the joint locations for each person and thus does not need a matching step.
>
> > The group matching is a little confusing, how do you select 'consecutive' joints in line 194?
>
> Fig.2 illustrates the group matching scheme. Suppose we have 150 joint queries (10 persons and 15 skeleton joints for each person), then the first 15 joint queries are grouped as a set for skeleton joints of the first person, the following 15 joint queries are grouped for the second person and so on, resulting in grouped joint queries for these 10 persons. We will add more details to illustrate our group matching scheme in revision.
>
> > how does the method handle occlusion settings?
>
> Our method addresses occlusion cases in three aspects: 1) It aggregates complementary multi-view information to address occlusion issues commonly existing in monocular settings. 2) Our method adopts the self-attention mechanism to reason over all skeleton joints, which helps alleviate the occlusion issues. For instance, when a skeleton joint is occluded, the model can still estimate its 3D location by reasoning over other skeleton joints that are not occluded. 3) Previous methods first estimate 2D poses or heatmaps on each view independently, which may suffer from occlusion due to monocular inputs, and result in poor 3D pose estimation. Differently, our MvP model directly regresses 3D joint position from the aggregated multi-view information and the learnable joint queries, and thus offers certain robustness  to occlusion issues.

---

> > ### Comment · Reviewer_vMiR · 2021-08-29
> > **post-rebuttal reviews**
> >
> > I am still confusing about the how do you decide the 'first 15 joints queries' belong to the same person? Or you randomly select 15 joints and no matter what, just treat them as the same person all 15 joints? If it is consecutive joints, how do you handle occlusion cases? or crowded cases? The question is also coupled with the hungarian during inference time. If you do not have hungarian, how do you do matching during inference?
> >
> > The reviewer Sx8n also comments with 'Could the proposed group matching handle some challenging cases, e.g., people hugging each other? The authors are encouraged to discuss more about the limits of the current joint-to-person association strategy.', which means there still limitations/future work with methods. Authors should put some efforts to write limitations/future work paragraphs.
> >
> > For the cross-dataset evaluation scenario raised by reviewer 6mm2 and Sx8n you may refer to
> > PoseAug: A Differentiable Pose Augmentation Framework for 3D Human Pose Estimation
> > or
> > Predicting Camera Viewpoint Improves Cross-dataset Generalization for 3D Human Pose Estimation
> > to solve that.
> > Authors are encouraged to discuss that.

---

> > > ### Author Response · Authors · 2021-09-02
> > > **Response to Reviewer vMiR**
> > >
> > > > I am still confusing about the how do you decide the 'first 15 joints queries' belong to the same person? Or you randomly select 15 joints and no matter what, just treat them as the same person all 15 joints? If it is consecutive joints, how do you handle occlusion cases? or crowded cases? The question is also coupled with the hungarian during inference time. If you do not have hungarian, how do you do matching during inference?
> > >
> > > The reviewer might have misunderstood the joint query grouping and Hungarian matching procedures. Here we give further clarification. Suppose there are 10 persons and 15 skeleton joints for each person. The proposed model is designed to have 150 joint queries. Every 15 joint queries are consecutively grouped in the joint query grouping procedure, as shown in Fig. 2. Each joint query is pre-set to represent a certain joint (e.g., the first query in each group corresponds to the nose, as shown in Fig. 2). The joint queries are trained together with the model to learn to localize the specific joints via the model’s internal attention mechanism.
> > >
> > > The output predictions are also arranged consecutively as the joint queries: the first 15 joint predictions correspond to joints of the first person and the first joint prediction corresponds to the first joint of the first person.
> > >
> > > The Hungarian matching procedure is ONLY used in the training procedure to find the optimal assignment between those grouped 3D pose predictions and the ground truth poses and compute the Hungarian loss correspondingly. For example, if there are 3 persons in the scene, i.e., 3 ground truth poses, we then find the best matched 3 poses from the 10 grouped poses to the ground truth poses via Hungarian matching and compute the Hungarian loss with the obtained optimal assignment.
> > >
> > > The group matching is NOT needed during inference. We directly group every consecutive 15 joints as a pose prediction, the confidence scores of the 15 joints are averaged as the final confidence score of the pose prediction. The model only outputs the predictions with pose confidence scores above a predefined threshold.
> > >
> > > Our model is not designed to pay extra attention to the occlusion and crowded cases which are out of the scope of this work. But it should work well in these challenging cases as the model can effectively aggregate multi-view information via its internal attention-based information aggregation. The joints occluded from a specific view would be visible from some other views and the proposed model would leverage such information to predict the occluded joints.
> > >
> > >
> > >
> > > > The reviewer Sx8n also comments with 'Could the proposed group matching handle some challenging cases, e.g., people hugging each other? The authors are encouraged to discuss more about the limits of the current joint-to-person association strategy.', which means there still limitations/future work with methods. Authors should put some efforts to write limitations/future work paragraphs.
> > >
> > > We agree that the cases of people hugging each other are challenging for existing pose estimation models (including ours). Since sufficiently solving these challenging cases are out of scope of this work, we did not explicitly discuss how our model can handle these cases. But as mentioned in the response to Reviewer Sx8n, if the training data capture such challenging cases, our MvP model can learn to differentiate different persons and accurately estimate their 3D poses by aggregating the multi-view information. This is possible because our model can directly predict the 3D joint locations, getting rid of the errors incurred in the intermediate steps (like 2D pose estimation). Besides, the group matching strategy can find optimal matching between the ground truth poses and the pose predictions, and provide accurate supervision signals during training. Thus, our model is potentially robust to the challenging occlusion.
> > >
> > > Our model currently does not have specific designs to address the more challenging crowded scenarios. But it is interesting to explore along this direction, like employing some one-to-many assignment strategy to enable prediction of spatially very close subjects (e.g., persons hugging together) with the same set of joint queries, motivated by some existing approaches for crowd detection [R1, R2].
> > >
> > >
> > > The main limitation of MvP lies in cross-view/dataset generalization since we train the model on indoor dataset with fixed camera viewpoints. We could disentangle the multi-view feature learning and camera parameters as well as training the model on both the indoor and the synthetic datasets jointly to address this limitation as discussed in the response to reviewer 6mm2. We could also explore the strategies presented by PoseAug [R3] to improve the model’s cross-view/dataset generalization. The details are given in the following answer.
> > >
> > > The above discussion on crowd/occlusion scenarios and limitations will be incorporated into revision.
> > >
> > >
> > > [R1] Detection in Crowded Scenes: One Proposal, Multiple Predictions, Conference on Computer Vision and Pattern Recognition.
> > >
> > > [R2] Detecting and Matching Related Objects with One Proposal Multiple Predictions. Conference on Computer Vision and Pattern Recognition Workshop.
> > >
> > > [R3] PoseAug: A Differentiable Pose Augmentation Framework for 3D Human Pose Estimation, Conference on Computer Vision and Pattern Recognition.
> > >
> > >
> > > > For the cross-dataset evaluation scenario raised by reviewer 6mm2 and Sx8n you may refer to PoseAug: A Differentiable Pose Augmentation Framework for 3D Human Pose Estimation or Predicting Camera Viewpoint Improves Cross-dataset Generalization for 3D Human Pose Estimation to solve that. Authors are encouraged to discuss that.
> > >
> > > The evaluation settings from the suggested works [R3, R4] are not suitable for evaluating our multi-view multi-person model’s cross-dataset generalization, as they are designed for  single-person and single-view datasets (Human3.6M, MPI-INF-3DHP and 3DPW). To evaluate our models under cross-dataset scenarios, we can build a multi-view multi-person synthetic dataset by extending image simulators [R5, R6] to multi-view multi-person data generation. Then we can use this synthetic dataset as well as the Panoptic dataset for evaluating models' cross-dataset generalization, i.e., we can train our model on the Panoptic dataset and evaluate it on the synthetic dataset and vice versa. We will explore this in the future.
> > >
> > > In the future, we will explore how to improve the model generalization to unseen scenarios that are very different from the training ones. One possible solution is to train our MvP model on both the indoor and the synthetic datasets jointly. Following the idea of PoseAug, we could also implement the image simulators [R5, R6] as data augmentor to generate extra multi-view multi-person data for model training. The data augmentor can be trained (as PoseAug [R3]) to control the simulator parameters to generate augmented data with different person postures, body shapes, appearances and viewpoints by taking the training loss from the MvP model as feedback signals in an online manner similar to PoseAug [R3]. Training MvP on these augmented data with more diverse distribution can improve its generalization to cross-view/dataset scenarios.
> > >
> > > [R3] PoseAug: A Differentiable Pose Augmentation Framework for 3D Human Pose Estimation, Conference on Computer Vision and Pattern Recognition.
> > >
> > > [R4] Predicting Camera Viewpoint Improves Cross-dataset Generalization for 3D Human Pose Estimation, European Conference on Computer Vision Workshop.
> > >
> > > [R5] Learning to Train with Synthetic Humans, Conference on German Conference on Pattern Recognition.
> > >
> > > [R6] AGORA: Avatars in Geography Optimized for Regression Analysis, Conference on Computer Vision and Pattern Recognition.

---

### Decision · Program_Chairs · 2021-09-27

**Decision:**

Accept (Poster)

**Comment:**

Reviewers agreed this is an interesting paper and assigned scores ranging from 5 to 8. The rebuttal successfully clarified some of the key reviewers’ initial concerns and the ACs reached consensus that this paper can be accepted for publication. Authors are highly encouraged to address the key comments reported by reviewers in the final camera-ready version